# Punctuated evolution of canonical genomic aberrations in uveal melanoma

Matthew G. Field[1], Michael A. Durante[1], Hima Anbunathan[2], Louis Z. Cai[1], Christina L. Decatur[1], Anne M. Bowcock [2], Stefan Kurtenbach[1] & J. William Harbour [1]

Cancer is thought to arise through the accumulation of genomic aberrations evolving under Darwinian selection. However, it remains unclear when the aberrations associated with metastasis emerge during tumor evolution. Uveal melanoma (UM) is the most common primary eye cancer and frequently leads to metastatic death, which is strongly linked to *BAP1* mutations. Accordingly, UM is ideally suited for studying the clonal evolution of metastatic competence. Here we analyze sequencing data from 151 primary UM samples using a customized bioinformatic pipeline, to improve detection of *BAP1* mutations and infer the clonal relationships among genomic aberrations. Strikingly, we find *BAP1* mutations and other canonical genomic aberrations usually arise in an early punctuated burst, followed by neutral evolution extending to the time of clinical detection. This implies that the metastatic proclivity of UM is "set in stone" early in tumor evolution and may explain why advances in primary treatment have not improved survival.

[1] Bascom Palmer Eye Institute, Sylvester Comprehensive Cancer Center and Interdisciplinary Stem Cell Institute, University of Miami Miller School of Medicine, Miami, Florida 33136, USA. [2] National Heart and Lung Institute, Imperial College London, London, SW3 6LR, UK. Correspondence and requests for materials should be addressed to J.W.H. (email: harbour@miami.edu)

Uveal melanoma (UM) is the most common cancer of the eye and leads to metastatic death in up to half of patients. UMs are commonly divided into two prognostic subgroups based on their gene expression profile (GEP); those with the class 1 GEP have a low risk of metastasis, whereas those with the class 2 GEP have a high risk[1]. UM is also notable for having two sets of driver mutations, with each tumor typically containing one mutation from each group[2]. One group consists of mutually exclusive gain-of-function mutations in members of the Gq signaling pathway—GNAQ, GNA11, CYSLTR2, or PLCB4. These "Gq mutations" are present in virtually all UMs, are not prognostically significant, are insufficient alone to induce malignant transformation, but are seemingly required to initiate tumorigenesis[3–8]. The other group consists of near-mutually exclusive mutations in BAP1, SF3B1, and EIF1AX ("BSE" mutations), which are strongly prognostic of metastatic risk. Inactivating mutations in the tumor suppressor BAP1 are associated with the class 2 GEP and high metastatic risk[9], whereas single nucleotide substitutions in SF3B1 and EIF1AX are found mainly in class 1 tumors and are associated with intermediate and low metastatic risk, respectively[10,11]. UMs are also associated with a small set of recurrent chromosome copy number alterations (CNAs), which tend to occur in the context of a specific GEP class and BSE mutation[12]. Loss of heterozygosity for chromosome 3 (LOH3) is frequently found in BAP1-mutant class 2 tumors and is thought to represent the "second hit" in the bi-allelic loss of BAP1, located at chromosome 3p21[9]. 6p Gain (6p+) is often found in class 1 tumors harboring SF3B1 and EIF1AX mutations, whereas 8q gain (8q+) can be found in both class 1 and class 2 tumors, and is associated with BAP1 and SF3B1 mutations.

Despite success in identifying these canonical genomic aberrations in UM, how and when these events arise during tumor evolution remains unknown. In cutaneous melanoma, progression from benign nevus to malignant melanoma follows a typical Darwinian model of gradual evolution[13], characterized by successive waves of mutations, clonal expansions, and selective sweeps fueled by ultraviolet radiation-induced DNA damage, with CNAs occurring relatively late[14,15]. An obstacle to performing reliable genomic clonality analysis in UM has been an inability to detect the wide diversity of BAP1 mutations using standard bioinformatic methods. Here we analyze next generation

sequencing (NGS) data from 151 primary UMs using a wide range of bioinformatic tools and techniques to optimize our detection of BAP1 and other mutations and CNAs, to explore their clonal relationships. This approach reveals many previously undetected BAP1 and spliceosome mutations, and uncovers strong evidence that the canonical genomic aberrations in UM usually arise in an early, punctuated burst followed by clonal stasis. These findings underscore the striking differences in genomic structure and evolution between UM and cutaneous melanoma, and they have profound implications for treatment and survival in UM.

## Results

**Data sets for genomic analysis.** We initially analyzed whole exome sequencing (WES) data from 139 primary UM samples, including 37 from the practice of the senior author (J.W.H.), 80 from the The Cancer Genome Atlas (TCGA), and 22 from a publicly available data set (UNI-UDE)[11] to identify driver mutations and CNAs (Fig. 1, Supplementary Table 1, and Supplementary Data 1). GEP classification data were available for all JWH samples and was estimated using RNA sequencing (RNA-seq) data for TCGA samples[16].

**Detection of BAP1 mutations.** Recent analyses have found BAP1 mutations in only about 20% of UMs[11,17], yet ~ 40% of UMs are expected to harbor these mutations[9]. Thus, we suspected that BAP1 mutations may frequently be missed when they comprise large insertions/deletions (indels), intronic/splicing alterations, and other complex rearrangements of the BAP1 locus[9]. To enhance our ability to detect such a wide range of inactivating mutations, we developed a robust pipeline of complementary bioinformatic tools to improve read alignments, manage low read counts, and identify large genomic alterations (Supplementary Fig. 1). In comparison with the Firehose analysis[17], which detected BAP1 mutations in 17 (21%) of the 80 TCGA samples using the GATK pipeline (Supplementary Fig. 2), our pipeline detected BAP1 mutations in 36/80 (45%) of the same samples (Supplementary Data 1). Similarly, Martin et al.[11] identified BAP1 mutations in 4 (18%) of 22 samples, whereas we identified BAP1 mutations in 9 (41%) of the same samples (Supplementary Data 1). Overall, BAP1 mutations were detected in 62 (45%) of

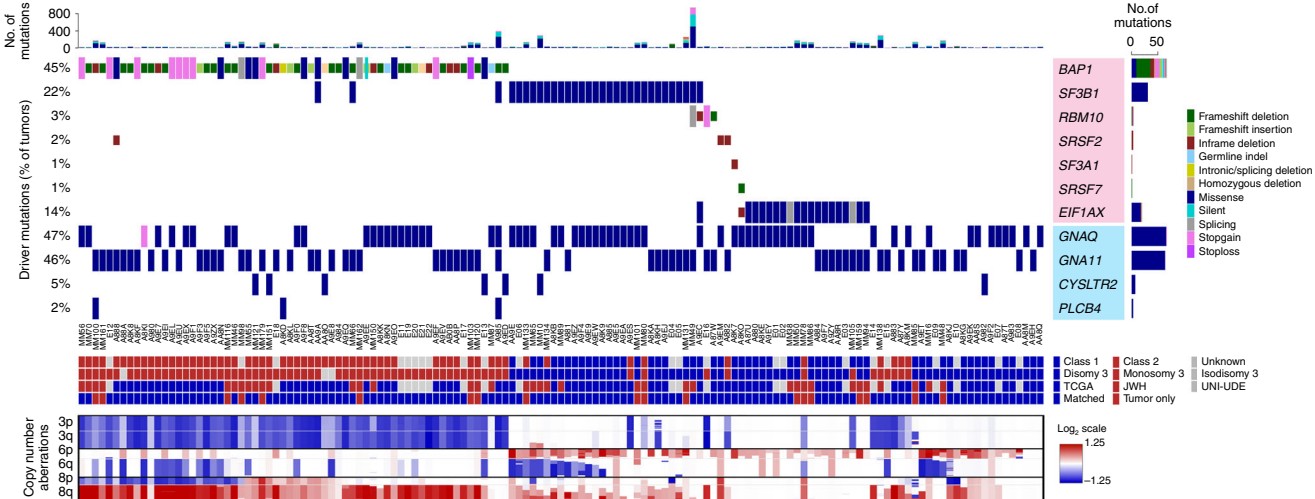

**Fig. 1** Molecular landscape in 139 primary uveal melanomas analyzed by whole exome sequencing. Mutation status for common driver and spliceosome mutations, type of mutation, common chromosome copy number alterations (CNAs), gene expression profile status (class 1 versus class 2), source of tumor sample, and availability of matched normal DNA are indicated. CNAs were assessed using CNVkit[62]. CNA data are scaled using the log2 copy ratio of the predicted copy number over the normal copy number. Mutations in "BSE" and splicing genes (pink box) are demarcated from those in Gq signaling pathway genes (blue box)

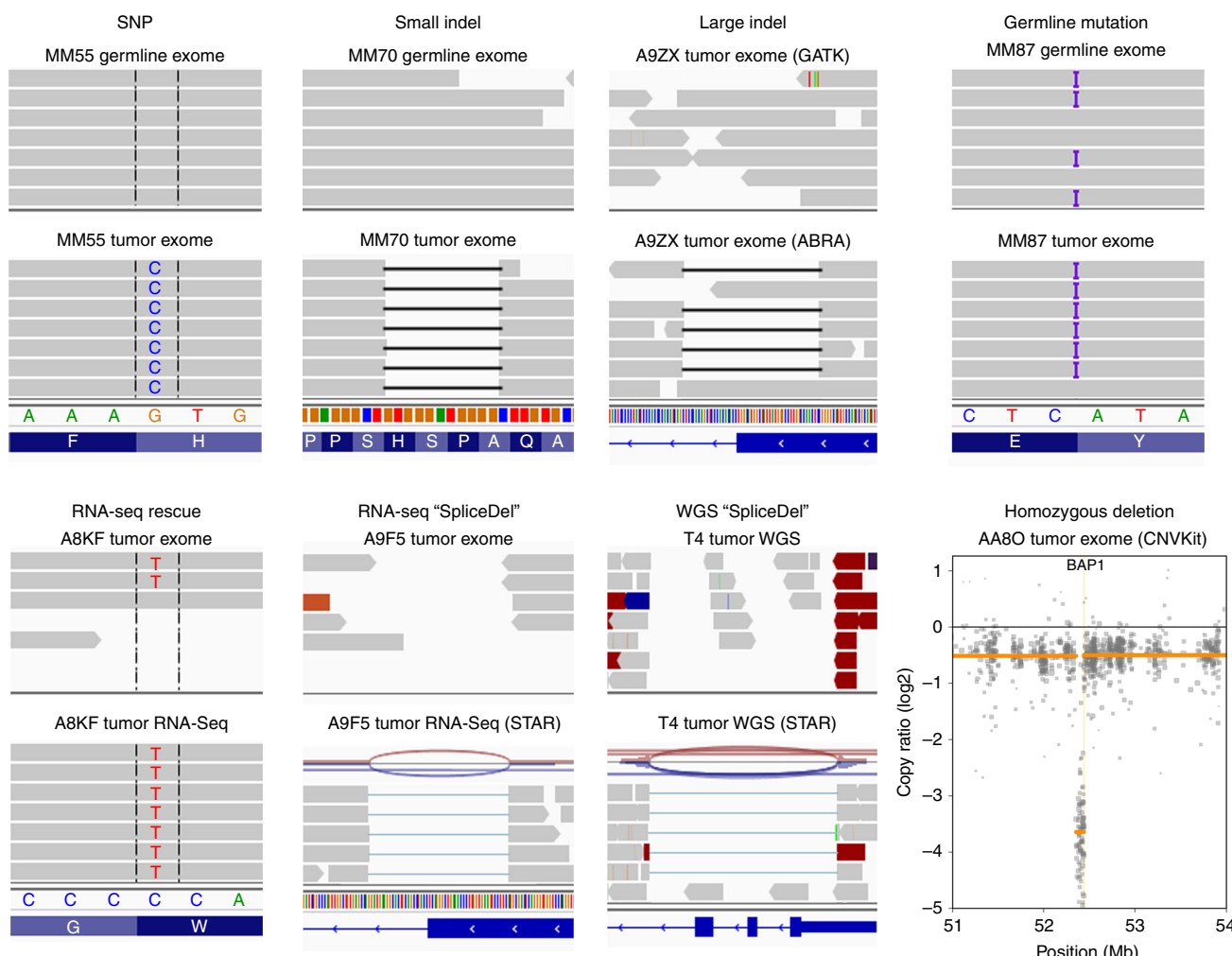

**Fig. 2** *BAP1* mutation diversity and necessary detection methods. Standard somatic mutation callers (i.e., Varscan2 and MuTect2) can detect SNPs (SNP) and small indels (small indel). Large indels need specialized realignment (i.e., with ABRA) for detection, as these can be missed with IndelRealigner and/or Mutect2 (large indel). Somatic mutation callers exclude germline mutations, so blood samples need to be run with a germline mutation caller (e.g., HaplotypeCaller) or as an unmatched "tumor" with a somatic caller (e.g., MuTect2) to detect patients with *BAP1* hereditary cancer predisposition syndrome (germline mutation). Mutations can be "rescued" in low coverage regions by combining WES and RNA-Seq data (i.e., UNCeqR) (RNA-seq rescue). Large indels may be missed with WES data when they start or extend into intronic or promoter regions due to poor bait coverage. In the case of the former, RNA-Seq data can be used to detect large indels that start in a transcribed exon and extend into the intron using alternative splicing algorithms (RNA-seq "SpliceDel"). For the latter, WGS with alternative splicing algorithms is required to detect indels that start in the promoter region or 5'-UTR and extends into exons or across multiple exons (WGS "SpliceDel"). In cases with loss of one chromosome 3 and deletion of genes on the remaining copy of chromosome 3, a CNA caller is required to detect the homozygously deleted regions (homozygous deletion). Sample alignments were visualized using Integrative Genomics Viewer. A representative sample was selected for each mutation type and detection method

the 139 WES samples queried in this study, including 9 non-synonymous single-nucleotide polymorphisms (SNPs), 10 stop-gain/stoploss SNPs, 2 intronic/splice SNPs, and 41 indels (Fig. 2 and Supplementary Data 1). The use of two different mutation callers (MuTect2 and Varscan2) improved detection of *BAP1* mutations, 17 of which were only called by one algorithm or the other. Using ABRA to re-align around large indels[18], we detected *BAP1* deletions of > 20 base pairs in 10 samples that were not previously detected by other methods (Fig. 2). In one sample (A8KF), we detected a *BAP1* mutation that was initially filtered out due to low exome coverage but was "rescued" with matched RNA-Seq data using UNCeqR[19] (Fig. 2). In addition, due to the design of baits for exome capture, large deletions that start or end in intronic or promoter regions can be missed using WES data. In three samples (AA8P, A9F5, and AB0B), we used RNA-seq data and the STAR aligner to identify large splicing deletions ("SpliceDels") that start in an exon and extend into an intron (Fig. 2).

These samples had low DNA coverage in the deleted regions, yet all three showed at least one read that called the deletion and multiple reads on either side of the deletion that did not span into the deleted region, corresponding precisely to the "SpliceDel" mutations detected by RNA-seq. In two samples (E21 and AA8O), we identified homozygous deletions of the *BAP1* locus using CNVkit that would be missed by most mutation-calling algorithms (Fig. 2). In two samples that were initially filtered out by our pipeline due to low coverage (A9EQ and E22), *BAP1* mutations were identified by visual inspection using the Integrative Genomics Viewer (IGV). A9EQ showed the same mutation in both WES and RNA-seq data, and E22 showed an obvious deletion in three out of six reads. Somatic mutation callers pair germline and tumor samples to filter out germline SNPs. However, as *BAP1* mutations are known to occur in the germline in some cases, we analyzed all blood samples independently of their matched tumor using MuTect2, with which we identified two

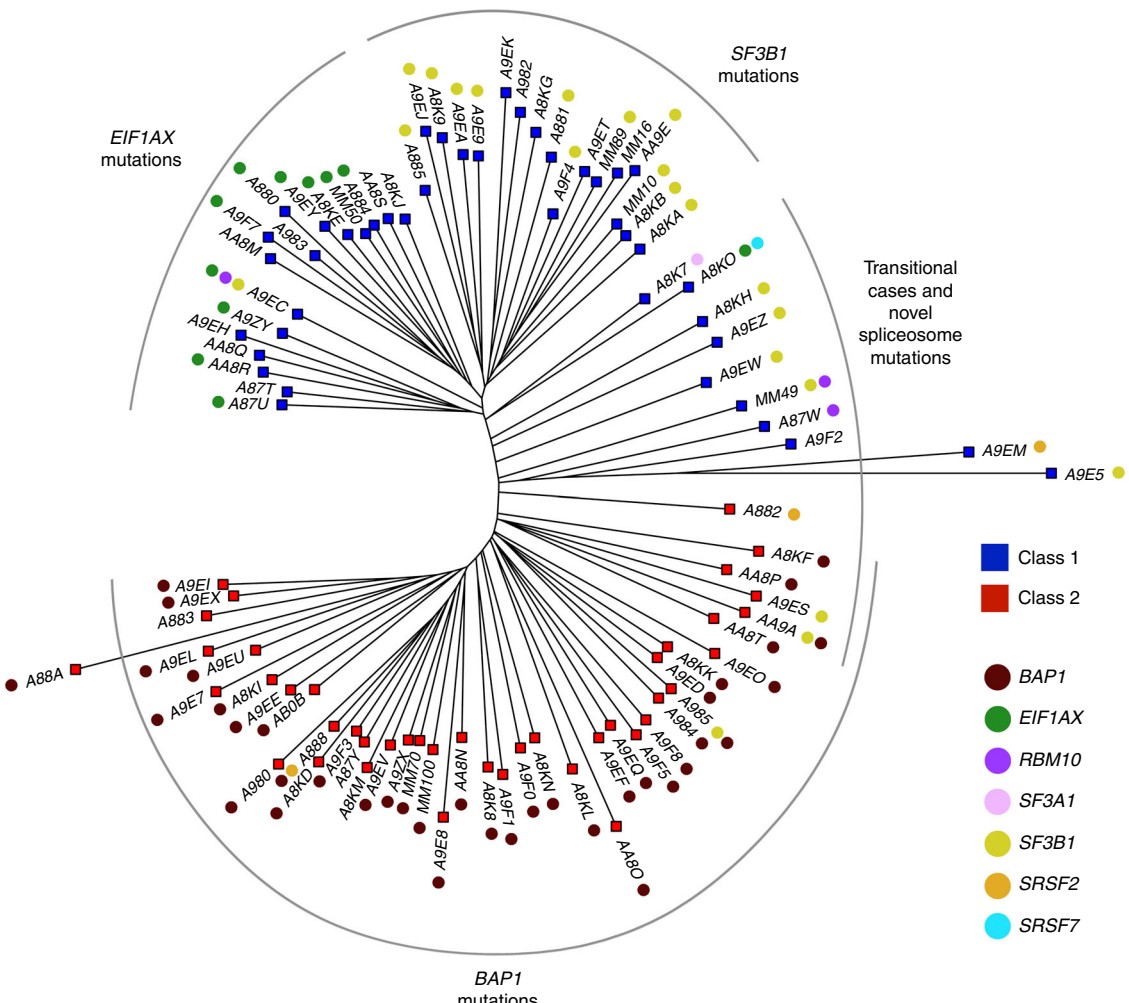

**Fig. 3** Phylogenetic tree demonstrating relationships between uveal melanoma samples based on genomic DNA methylation. A minimum evolution algorithm (Canberra distance) was used to build an unrooted evolutionary tree. Gene expression profile (class 1 versus class 2) and mutation status are indicated. These data are presented in a rooted phylogenetic tree with significance estimates in Supplementary Fig. 5

cases with germline *BAP1* mutations (MM87 and A8KN) (Fig. 2). Even with our optimized pipeline, we did not identify a *BAP1* mutation in 10 of the class 2 samples; thus, we suspected that some *BAP1* mutations are not detectable with WES or RNA-seq data, because they span multiple exons or start in untranscribed regions. Indeed, when we analyzed publicly available whole genome sequencing (WGS) data from 12 UM samples reported by Furney et al.[20] using our "SpliceDel" technique, we identified 2 samples containing multi-exonic *BAP1* deletions that started in the 5′-untranslated region and intronic regions (Fig. 2 and Supplementary Data 2), which were previously unreported. Hence, a complimentary repertoire of sequencing and analytical techniques will be required to detect all *BAP1* mutations.

**Detection of other driver and putative driver mutations**. Of the 139 WES samples analyzed, mutations in Gq pathway genes were detected in 137 (98.6%) samples and were mutually exclusive in all except two cases (Fig. 1 and Supplementary Fig. 3). *SF3B1* mutations were present in 31 (22%) samples and *EIF1AX* mutations were present in 20 (14%) samples (Supplementary Data 1 and Supplementary Fig. 3). Among samples without a detectable BSE mutation, five were found to harbor mutations in additional spliceosome factors, including *RBM10* (two samples), *SRSF2* (two samples), and *SF3A1* (one sample) (Supplementary Fig. 3). The *SRSF2* mutations consisted of deletions encompassing

the P95-R102 region previously reported to disrupt splicing in myelodysplastic syndrome[21]. We then reviewed the BSE-mutant samples and found additional mutations in *RBM10* (two samples), *SRSF2* (one sample), and *SRSF7* (one sample). Although the functional consequences of the SF3A1 and SRSF7 mutations are not known, the tumors containing these mutations cluster with the SF3B1-mutant tumors (Fig. 3), suggesting that they may confer similar functional consequences as SF3B1 mutations. Indeed, all of these spliceosome factors have been shown to be mutated in other cancers[22–25], and they interact functionally with each other and with SF3B1[26,27], which may explain the tendency for these mutations to be mutually exclusive with *SF3B1* mutations. Although cancer-associated alterations in SF3B1 are mostly change-of-function hotspot mutations[20], this is not the case for all splicing factors. For example, *RBM10* undergoes frameshifts, truncations, and indels in lung cancer similar to our findings in UM[28], suggesting that oncogenic mechanisms may vary among different splicing factor mutations.

**Mutational signature**. Using a probabilistic modeling algorithm to analyze the 139 WES samples[29], we identified three mutational signatures (Supplementary Fig. 4). The most prominent signature was a cytosine-to-thymine transition at CpG dinucleotides, which has been associated with aging[30]. This deamination event accounts for the mutation hotspot at codon 183 in *GNAQ* and

*GNA11*, as well as the mutation hotspot at codon 625 in *SF3B1*. The second most prominent signature was a cytosine-to-adenine transition, which has been associated with reactive oxygen damage[31]. Interestingly, when samples were analyzed separately based on their BSE mutation, this signature was enriched only in *SF3B1* mutant tumors. Thymine-to-guanine transversions were also enriched in all subgroups and are of unknown etiology. Consistent with other reports[5,32], we found no ultraviolet radiation signature. Notably, despite ostensible links between BAP1 and BRCA1[33], no DNA double-strand break-repair "BRCA signature" was detected in *BAP1*-mutant UMs or any other subgroup, calling into question whether there is in fact a functional link between BAP1 and BRCA1/2 in tumorigenesis[33].

**Methylomic clustering analysis**. Previous efforts to classify UMs have been based on CNAs[34] and GEP[35]. Here we used global DNA methylation profiling to infer inherent evolutionary relationships between tumors[36], based on the assumption that uveal melanocytes (from which UMs arise) will have similar methylomic profiles among different individuals. Methylation profiling has the advantage of being agnostic to mutation status and GEP. We used a minimum evolution algorithm[37] to analyze 87 samples in which global DNA methylation data were available, which revealed phylogenetic clusters that correlate strongly with GEP (class 1 versus class 2) and with BSE mutations (Fig. 3 and Supplementary Fig. 5). Thus, most UMs evolve along one of three canonical trajectories toward fitness maximums denoted by *BAP1*, *SF3B1*, or *EIF1AX* mutations. Most tumors containing alternative spliceosome mutations (*RBM10*, *SRSF2*, *SF3A1*, and *SRSF7*) lie between the *SF3B1* and *BAP1* clusters and may therefore represent intermediate transitional cases or uncommon non-canonical trajectories.

**Clonality analysis**. Bulk genomic sequencing data provide a chronicle of genomic aberrations that occur during tumor evolution, which can be used to infer the life history of a given tumor[38]. To reveal underlying patterns in the clonal architecture of UM, we analyzed WES data from the 80 TCGA samples and WGS data from the 12 Furney samples[20] using a hierarchical Bayesian Dirichlet process to identify clonal mutations and the cgpBattenberg algorithm to identify clonal CNAs (Supplementary Fig. 6)[38,39]. As expected, in almost all samples, an initiating Gq pathway mutation was present in 100% of tumor cells, indicating that it occurred before the appearance of the most recent common ancestor (MRCA)[38]. Henceforth, we focused on the BSE mutations due to their prognostic gravity, anticipating that they would map to subclones that arose after the MRCA. Surprisingly, however, most samples contained BSE mutations and their associated CNAs (LOH3, 6p+, and 8q+) in 100% of tumor cells (Fig. 4a,b and Supplementary Fig. 7). In a small number of samples, all but one canonical aberration were present in the MRCA, with the other one in a subclone (Fig. 4a,b). For example, five class 1 tumors contained an *EIF1AX* mutation in the MRCA, with 6p+ arising in a later subclone. Similarly, LOH3 was always present in 100% of tumor cells, whereas the associated *BAP1* mutations were occasionally found in a subclone. In addition, most alternative spliceosome mutations (*RBM10*, *SRSF2*, *SF3A1*, and *SRSF7*) mapped to subclones. Hence, the canonical genomic aberrations in UM usually arise in an early punctuated evolutionary process[40-42], with little ongoing acquisition of new driver aberrations after the appearance of the MRCA.

**Evidence for neutral tumor evolution**. As our findings suggested that all of the genomic events necessary for the malignant phenotype in UM arise before or soon after the emergence of the MRCA, we postulated that most or all additional mutations accumulating after the appearance of the canonical genomic aberrations are evolutionarily neutral[14,42]. To test this hypothesis, we used a power-law distribution model to assess the likelihood of neutral tumor growth in WGS data from the 12 Furney samples[20], as the small number of mutations found in WES data did not allow for proper modeling (Fig. 5a). All 12 samples showed a goodness-of-fit ($R^2$) well above the 0.98 threshold for neutral tumor evolution (Fig. 5b). Indeed, the UM samples fit the neutral evolution model better than most other cancer types that have been evaluated (Fig. 5c)[14]. These findings support a punctuated evolution model in which the small handful of canonical aberrations necessary to reach a fitness maximum occur early, beyond which there is little ongoing clonal selection or adaptation in the primary tumor.

**Discussion**

Historically, cancer has been thought to develop through the gradual accumulation of numerous mutations over long periods of time, with occasional "driver mutations" giving rise to new subclones that evolve under ongoing selective pressure[13-15]. Consistent with recent landmark studies that have challenged this model[40-42], our findings reveal punctuated rather than gradual evolution in UM. This finding is surprising, as most previous work suggested that CNAs occur in a successive manner in UM[43]. Most UMs harbor one Gq pathway mutation (*GNAQ*, *GNA11*, *CYSLTR2*, or *PLCB4*), one BSE mutation (*BAP1*, *SF3B1*, or *EIF1AX*), and a few recurrent CNAs, in 100% of tumor cells. Hence, these canonical aberrations usually occur relatively early, before the emergence of the MRCA, consistent with punctuated evolution[40-42]. Subsequent aberrations accumulate following a power-law distribution characteristic of neutral tumor evolution[14]. As most UMs cease to accrue subsequent driver mutations once they acquire a BSE mutation, these mutually exclusive aberrations appear to represent alternative fitness maximums, with mutation of one gene relieving selective pressure to mutate the others. Taken together, these findings imply that the metastatic proclivity of a given tumor, which is strongly associated with its respective BSE mutation, may be "set in stone" early in tumor evolution, often before detection of the primary mass, which may explain the lack of improvement in survival rates despite advances in diagnosis and treatment[44].

Although the canonical aberrations arise through a punctuated evolutionary mechanism, they do not necessarily occur simultaneously. Our findings, taken together with previous work[3,4,7,9,10] suggest that Gq pathway mutations are early events required to initiate tumorigenesis but insufficient alone for malignant transformation, whereas BSE mutations confer malignant potential but are unable to trigger clonal expansion without a Gq mutation. Thus, BSE mutations either arise subsequent to Gq mutations or, if they occur first, they remain silent unless "unleashed" by a Gq mutation. The latter mechanism is presumably operative in patients with a germline *BAP1* mutation, two of which were included in this study. In both cases, a Gq mutation was present in 100% of tumor cells but was absent in the germline. Hence, a preexistent *BAP1* mutation was present in all uveal melanocytes but clonal expansion only occurred after one of these cells acquired a Gq mutation. This need for a tumor-initiating Gq mutation could explain why only a minority of individuals with germline *BAP1* mutations develop UM[45]. It is also interesting that several samples analyzed for clonality harbored more than one BSE mutation. AA9A, A985 and T11 contained *BAP1* and *SF3B1* mutations, whereas A9EC contained *SF3B1* and *EIF1AX* mutations. As all of these BSE mutations were present in the MRCA, they most likely coexisted in the same tumor cells rather than in

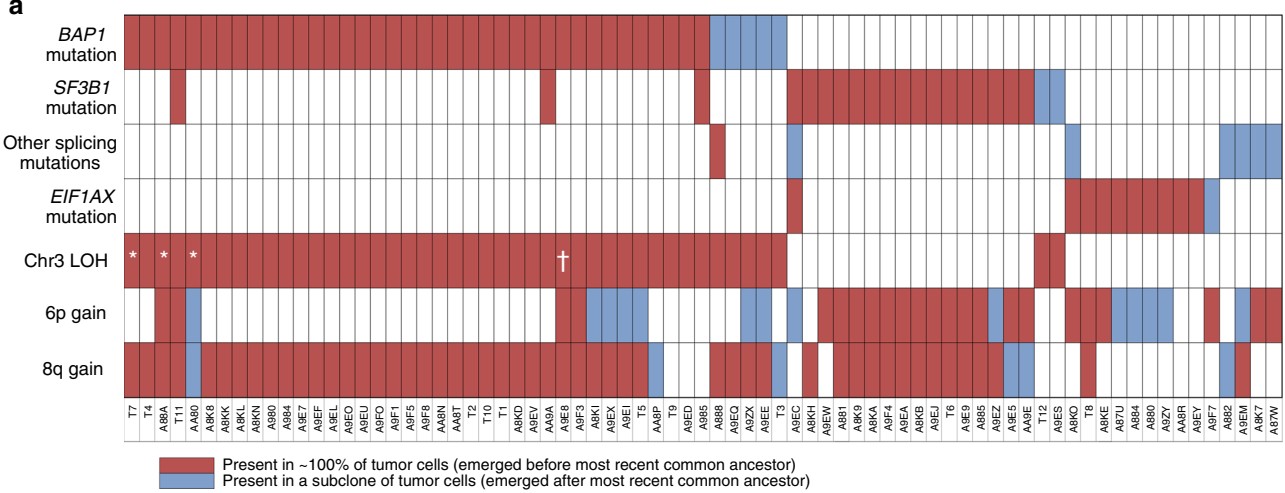

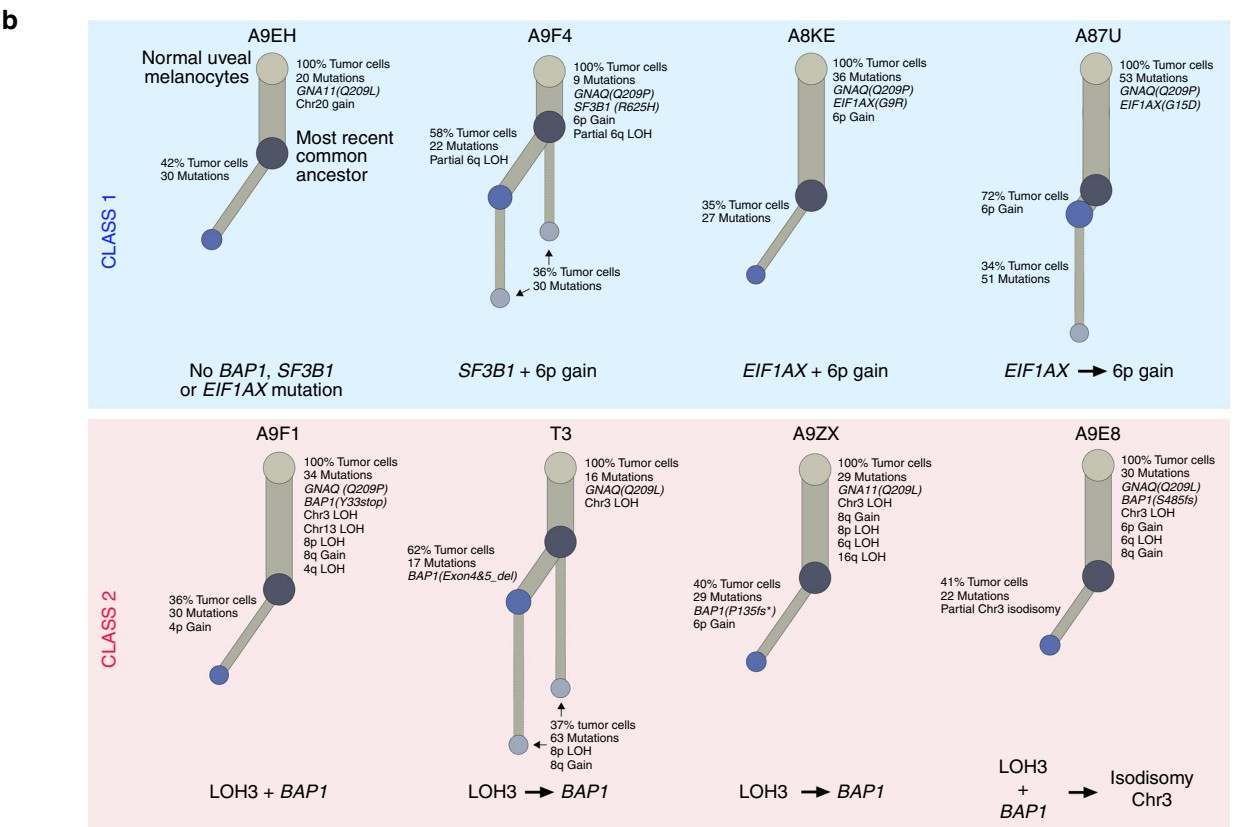

**Fig. 4** Clonal evolution in uveal melanoma. **a** Clonal relationships between canonical driver mutations and chromosome copy number alterations. Red bars denote aberrations that are present in 100% of tumor cells and thus arising before the most recent common ancestor (MRCA). Blue bars denote aberrations present in a subclone of <100% tumor cells arising subsequent to the MRCA. A white bar denotes that the aberration was not present. *Isodisomy 3 in 100% of tumor cells; †isodisomy 3 in a subclone of <100% of tumor cells. **b** Life history clonal evolution plots for eight representative uveal melanoma samples. Blue box illustrates class 1 tumors with no BSE mutations (A9EH), *SF3B1* and 6p gain in 100% of tumor cells (A9F4), *EIF1AX* and 6p gain in 100% of tumor cells (A8KE), and *EIF1AX* in 100% of tumor cells and 6p gain in a subclone (A87U). Red box illustrates class 2 tumors with LOH3 and *BAP1* in 100% of tumor cells (A9F1), LOH3 in 100% of tumor cells, and *BAP1* mutation in a subclone (T3 and A9ZX), and LOH3 and *BAP1* in 100% of tumor cells with later duplication of chromosome 3 to generate isodisomy 3 in a subclone (A9E8). The node at the top of each plot (light grey circle) represents normal uveal melanocytes, the presumed precursor cell that gives rise to all uveal melanomas. The next lower node (dark gray circle) represents the most recent common ancestor (MRCA) from which all aberrations present in 100% of tumor cells arose. Lower nodes (blue circles) represent subclones that arise after the most MRCA, with the area of the node being proportional to the percentage of cells in the subclone. The mutations and CNAs that occur between two nodes are indicated beside the connecting branch (gray bar). The length of each branch is proportional to the number of mutations that occur between nodes and the width of the branch is proportional to the percentage of cells containing those mutations. When a node contains rare aberrations that cannot be accurately mapped to one node, it is mapped to all possible nodes, as described in Methods

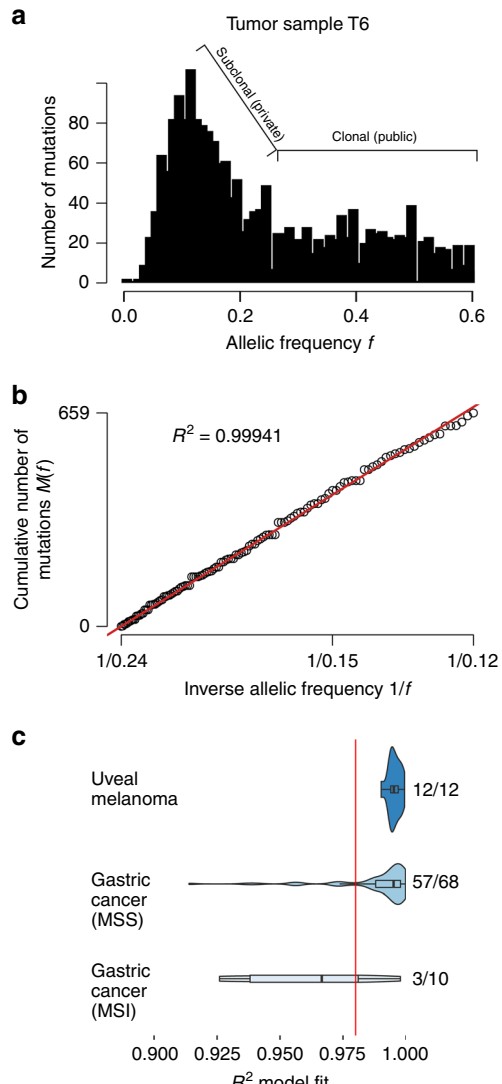

**Fig. 5** Analysis of uveal melanoma whole genome sequencing (WGS) data for evidence of neutral tumor evolution. **a** Histogram of mutant allele frequency for representative tumor sample T6. Mutations that occur at frequencies ≥ 0.25 are considered to be clonal (public) and those occurring at frequencies below this cutoff are subclonal (private)[14]. **b** Cumulative distribution, $M(f)$, of subclonal mutations for sample T6, which is highly consistent with the neutral evolution model[14]. **c** Comparison of goodness-of-fit, $R^2$, for neutral tumor evolution in uveal melanoma versus previously published gastric cancer WGS data[14]. All 12 uveal melanoma samples demonstrated $R^2 \geq 0.98$ (red line), whereas 60 of 78 (76.9%) gastric cancer samples without microsatellite instability (MSS) and only 3/10 (30%) of gastric cancer samples with microsatellite instability (MSS) met this threshold. The proportion of samples with a particular $R^2$ model fit is proportional to the violin plot width. Box plots within the violin plot illustrate the median, upper, and lower quartiles, and Tukey's whiskers (median ± 1.58 times interquartile range)

different tumor subclones. Interestingly, the GEP in these cases reflected the mutation associated with the worse prognosis.

Although most class 2 tumors contained a *BAP1* mutation and LOH3 in the MRCA, the *BAP1* mutation (but not LOH3) was occasionally present in a subclone, suggesting that LOH3 can precede mutation of *BAP1* on the other chromosome 3. 6p+ Was usually present in 100% of tumor cells in *SF3B1*-mutant tumors but only in a subclone in *EIF1AX*-mutant and *BAP1*-mutant tumors; 8q+ was usually present in 100% of tumor cells in *SF3B1*-

mutant and *BAP1*-mutant tumors but was rarely present in *EIF1AX* tumors. 8q+ In class 2/*BAP1*-mutant tumors usually consisted of multiple extra copies of the entire q arm, whereas in class 1/*SF3B1*-mutant tumors, it usually involved simple gain of a smaller fragment of 8q. There was no "smallest common region" of amplification to implicate a specific gene(s) under selective pressure. Taken together, CNAs appear to evolve along the evolutionary trajectory of their associated BSE mutation. Future studies are warranted to further delineate the early events in UM evolution in greater detail.

Of all the canonical aberrations in UM, the single most prognostically significant is the bi-allelic inactivation of *BAP1*, which is tightly linked to the class 2 GEP prognostic signature and high metastatic risk[9]. As such, we developed an enhanced bioinformatic pipeline to discover additional *BAP1* mutations. We show here that optimal detection of *BAP1* mutations requires a variety of different bioinformatic techniques, including special realignment of reads around large indels, use of multiple mutation callers, detection of homozygous deletions, and "SpliceDel" analysis of RNA-Seq and WGS data, to detect deletions spanning multiple exons or encompassing untranscribed regions. These findings indicate that the frequency of *BAP1* mutations in UM and other cancer types may be considerably underestimated. As expected, we did not observe an increased frequency of detecting other canonical mutations since they are mostly SNPs that are easily detected.

This study discloses a wide variety of previously unreported BAP1 and spliceosome mutations, and identifies punctuated evolution as an organizing principle behind the genetic, genomic, and transcriptomic landscape of UM. The early emergence of prognostically significant BSE mutations could explain why micrometastasis frequently occurs before diagnosis[46], and why more aggressive primary tumor treatment has not resulted in improved survival[44]. Consequently, these findings support an intensified effort to develop more effective treatments for metastatic disease.

## Methods

**Patients and sample collection.** Tumor and blood samples from the JWH data set were obtained from patients in the practice of the senior author who were diagnosed with UM arising from the choroid and/or ciliary body, and treated by primary enucleation without previous radiotherapy. The study was approved by the Institutional Review Board at the University of Miami and written informed consent was obtained from each patient. Clinical and histopathologic information were obtained and de-identified for further analyses. WES was performed on 40 tumor samples, 21 of which had matched blood DNA samples available for sequencing. DNA was extracted using the Wizard Genomic DNA Purification kit (Promega, Madison, WI) and the Quick Gene DNA whole blood kit S (Fugifilm, Tokyo, Japan), respectively. Exome fragments were captured using NimbleGen SeqCap EZ Human Exome Library v2.0 (Roche Nimblegen) and sequenced on the Illumina Genome Analyzer II (GAIIx). RNA from these cases was isolated using the PicoPure kit and sent to Castle BioSciences, Inc. for GEP to determine class 1 versus class 2 status[47].

**NGS data sources, quality control, and alignment.** Raw data files from 80 TCGA UM samples was obtained from the Cancer Genomics Hub (CGHub). Raw data files from the Martin et al.[11] and Furney et al.[20] raw data were obtained from the European Genome-phenome Archive (EGA). Data sets provided in BAM format (TCGA; Furney et al.[20]) were converted back into FASTQ files using bamUtil: bam2FastQ (v1.0.13) and were adjusted into proper format with unique read names using CGAT (v0.2.4)[48]. FASTQ files from all sources, including our data, then underwent the following bioinformatics pipeline. Sequence data were quality controlled using FastQC (v0.11.3). WES and WGS reads were trimmed (if required) and aligned to the human genome (hg19/GRCH37) using Novoalign (v3.04.06), marked for duplicates using Picard (v1.128), realigned around small and large indels using ABRA (v0.94c)[18], and read mate fixed and analyzed for coverage statistics using Picard. Tumor samples that had < 30 x coverage were excluded from further analysis. Unknown or unplaced contigs and mitochondrial genes were excluded from analysis. Raw RNA-seq FASTQ files from TCGA were obtained from CGHub and from Furney et al.[20] via EGA, assessed for quality using FastQC, and aligned to the genome using STAR (v2.5)[49]. For research purposes, GEP class

status of TCGA samples was estimated using the following method[16]. RNA-seq normalized count files were generated using DESeq2 in R[50]. Unsupervised principal component analysis (PCA) was conducted on the top 20% most variable expressed genes and plotted three-dimensionally using the stats and rgl packages in R, which grouped the samples into two clusters, as has been previously described for Class 1 and Class 2 tumors[47]. The identity of each cluster was determined to be most consistent with Class 1 versus Class 2 based on the expression of genes previously known to be differentially upregulated in each Class. Class assignment showed 100% concordance with the 11 TCGA samples in which Class status was determined using the clinically available DecisionDx-UM test. This method was used solely for research purposes and is not meant for actual clinical testing, as it has not been prospectively validated in a manner analogous to the DecisionDx-UM test.

**NGS mutation calling**. WES data sets underwent variant calling for SNPs and Indels using Varscan2 (v2.4.1)[51] and MuTect2 (GATK 2016-01-25 nightly build)[52]. Combining the two mutation callers was based on the finding that Varscan2 detects more true somatic and high-confidence SNPs than MuTect, whereas MuTect is better at detecting low-coverage SNPs.[53] MuTect2 is an updated SNP and indel mutation caller available from GATK that combines the original MuTect with the assembly-based machinery of HaplotypeCaller. MuTect2 has its own incorporated local realignment that adjusts for tumors with purity less than 100%, multiple subclones, and/or copy number variation (either local or aneuploidy). Varscan2 output was further refined using bam-readcount and a false positive filter (fplfilter)[54] with filter parameters set based on the TCGA-ICGC DREAM-3 SNV Challenge results, with adjustment of minimum average read trim length. For MuTect2, mutation calls for a panel of normals ($n = 117$ germline blood samples) was generated and pooled together to further filter out mutations that were present in at least two normals. For tumors without matched blood samples, MuTect2 was used for variant calling with a panel of normals and a high coverage blood sample to filter out likely germline mutations. MuTect2 was utilized for mutation calls of the WGS samples. To assess whether any mutations were missed by our pipeline due to low coverage, tumor-blood matched WES data were analyzed in combination with matched RNA-Seq data from the tumor using UNCeqR (v.0.2)[19] with default settings. In addition to the default settings for UNCeqR, mutations were filtered if both DNA and RNA alternate tumor read counts were < 3, < 20% of the total tumor read count, or the gene fell within the blacklist suggested by Radia[55]. For mutations called by Varscan2 or MuTect2, mutations were further filtered out if both alternate tumor read counts were < 3 and the minor allele frequency (MAF) was < 20%. For all sequencing samples, the BAM files were investigated manually for the regions of interest (*BAP1*, *SF3B1*, *EIF1AX*, *GNAQ*, *CYSLTR2*, *PLCB4*, and *GNA11*) using the IGV (v2.3.80, Broad Institute, Cambridge, MA). STAR-aligned RNA-seq data from the TCGA samples was also evaluated for "alternative splicing" within BAP1 exons, indicative of deletions not detected in the corresponding WES data due to the design baits for exome capture. The same strategy was used on the Furney et al.[20] WGS data set in which all regions of the genome are covered.

For all called mutations, annovar was used for annotation[56]. Following annotation, mutations were filtered out if the MAF was 1% or greater in the population according to 1000 Genomes Project (2015 August) or Exome Sequencing Project (2015 March), and mutations listed in dbSNP (v138) were filtered out, with the exception of SNPs with a MAF < 1% (or unknown) in the population, a single mapping to reference assembly, or with a "clinically associated" tag. Functional consequences of SNPs were assessed by three mutational predictor tools: Polyphen2 (probably damaging, possibly damaging, and benign)[57], FATHMM (damaging and tolerated)[58], and MetaLR (damaging and tolerated)[59]. In non-exonic regions and for synonymous mutations, SNPs were considered deleterious if two out of three of the above prediction algorithms predicted a damaging mutation. For non-synonymous exonic SNPs, mutations were not considered deleterious if two out of three algorithms predicted a benign or tolerated mutation. Probably damaging, possibly damaging, or damaging were considered deleterious calls. All insertions and deletions in exonic regions and alterations in splicing junctions were considered deleterious. A summary of the driver mutations, GEP status (class 1 versus class 2), source of tumor sample, and availability of matched normal DNA were mapped in a co-occurrence of mutations plot using ComplexHeatmap[60]. Lollipop plots displaying the distribution of driver mutations along the protein domains of each gene was plotted using MutationMapper (http://www.cbioportal.org/mutation_mapper.jsp) and domain information was populated based on default annotations in combination with a literature review[61].

After mutation calling, mutational signature analysis and visualization was conducted on WES data using the pmsignature package in R, which infers mutational signatures based on probabilistic models, similar to mixed-membership models used in population genetics and machine learning[29]. Signatures were modeled for nucleotide base changes with different combinations of ± 2 flanking bases using default parameters, which included a maximum of 10,000 iterations, a minimum of 10 iterations with different starting values, and a tolerance of estimation of 1e−04. Different numbers of mutational signatures were modeled, and the number of signatures was selected where the overall log-likelihood was highest without a rise in bootstrap error and where any given inferred signature did

not show a similar pattern across the genome to another inferred member. This analysis was conducted on all samples in combination, as well as on samples broken down into BSE subgroups.

**Copy number aberrations**. Copy number gains and losses were determined from WES and WGS data using CNVKit (v0.7.5, v0.7.10.dev0)[62] and by cgpBattenberg[38] for clonality analysis using default settings. Raw SNP 6.0 arrays from the TCGA data set and raw HumanOmni2.5 SNP arrays from the Furney et al.[20] data set were analyzed using ASCAT(v2.4)[63] with default settings. Isodisomy was determined by plotting MAF plots using CNVKit, ASCAT, TITAN[64], and cgpBattenberg.

**Phylogenetic analysis using DNA methylation data**. Raw output from the Infinium HumanMethylation450 BeadChip Kit (Illumina) was downloaded from TCGA ($n = 80$), and combined with our own cases ($n = 7$) and analyzed using the ChAMP R package[65]. The top 20% most variable methylation probe $\beta$-values were used for unsupervised PCA analysis using the stats and rgl R packages. Phylogenetic evolutionary models were generated using the minimum evolution algorithm (Canberra distance) in the ape R package[66]. Likelihoods of each bipartition in the tree were calculated using bootstrapping with 100 replicates.

**Clonality analysis**. To determine the clonality of copy number gains and losses, we used cgpBattenberg, which determines chromosome copy number gains and losses, and corrects for cell contamination of normal cells[38,67]. Adaptation of cgpBattenberg scripts and dependencies was performed by the High Performance Computing Core, Center for Computational Science at the University of Miami, for proper installation on the CentOS operating system used by the Pegasus supercomputer. Clusters of mutations were determined using a Bayesian Dirichlet process model that involves Gibbs sampling to estimate the posterior distribution of the parameters of interest. Mutations were assigned to the cluster in which they were most likely to fall based on variant allele frequency, following adjustment for normal cell contamination and copy number gains and losses using previously described algorithms and publicly available R scripts[38]. This methodology minimizes the risk of underestimating indel frequencies relative to SNPs by assuming that all mutations in the first cluster are in the MRCA. Two independent researchers reviewed the mutation cluster plots and counted the number of discrete clusters. When disagreement existed, the higher cluster number was selected. Subclones in which the mean of the cluster had a tumor read count < 10% of the allelic fraction were excluded as an independent subclone due to the possibility that these mutations could be spread across multiple different clones. Mutations attributed to this subclone were assigned to the next smallest cluster. For each cluster, the 95% confidence interval fitted for the Bayesian Dirichlet process model was used to calculate the number of mutations in the cluster. Subclonal copy number gains and losses, mutation clusters, and adjusted mutation variant allele frequency information were integrated using the pigeonhole principle to determine phylogenetic trees for each sample[38,67]. A custom python script was developed to automate generation of intra-tumor evolutionary trees, which takes inputs from the above analyses and generates an editable scalable vector graphics (.svg) file visualizing the life history, or phylogenetic tree, for a given sample. This algorithm can be provided upon request. Combining cgpBattenberg and the Bayesian Dirichlet process model for clonality analysis of CNAs and mutations, respectively, is the only publicly available methodology at this time for simultaneously determining the clonal evolution of CNAs and mutations occurring in the same genomic region. Other clonality algorithms are limited in that they either (1) only evaluate mutations in regions without CNAs or (2) pair mutations with CNAs in regions with CNAs and assume they occur together.

**Neutral tumor evolution**. To assess whether the evolution of UM fits a Darwinian versus a neutral tumor evolution model, we analyzed the 12 Furney et al.[20] WGS samples using a simple power-law distribution model predicted by neutral growth[14]. Mutations with variant allele frequencies between 12% and 24% were selected to account only for reliably called subclonal mutations and tumor purity in the samples. Proper modeling required samples to have a minimum of 12 mutations within this fitness boundary[14]. WES samples were not used for this analysis because none of them met this criterion. A custom script was provided by Williams et al.[14]. A goodness-of-fit ($R^2$) value of greater than 0.98 was considered to positively fit a neutral tumor evolution model. Calculated $R^2$ values for UM WGS cases were compared to $R^2$ values from WGS gastric cancer cases and plotted in a violin plot using ggplot2 in R.

**Data availability**. All sequencing and methylation array data generated from the practice of the senior author (J.W.H.) have been deposited in and are available from the dbGaP database under dbGaP accession phs001421.v1.p1.

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

## Acknowledgements

We are grateful to the patients who generously contributed samples for this research. We thank Lisle Mose and Eric Talevich for their assistance with the ABRA and CNVkit programs, respectively. We acknowledge the support of the Biostatistics & Bioinformatics and Oncogenomics Shared Resources at the Sylvester Comprehensive Cancer Center, and the University of Miami Center for Computational Science. The data used here were generated in part by the TCGA Research Network, Cancer Research UK Manchester Institute, Institut Curie, and University of Duisburg-Essen. This work was supported by National Cancer Institute grants R01 CA125970 (J.W.H.), R01 CA161870 (J.W.H. and A. M.B.) and F30 CA206430 (M.G.F.), Research to Prevent Blindness, Inc. Senior Scientific Investigator Award (J.W.H.), Melanoma Research Foundation (J.W.H. and M.G.F.), Melanoma Research Alliance (J.W.H.), Ocular Melanoma Foundation (J.W.H.), the 2015 RRF/Kayser Global Pan-American Award (J.W.H.), the Alcon Research Institute (J.W. H.), the Sylvester Comprehensive Cancer Center (J.W.H.), the University of Miami Sheila and David Fuente Graduate Program in Cancer Biology (M.G.F. and M.A.D.), the Center for Computational Science Fellowship (M.G.F. and M.A.D.), the AACR-Ocular Melanoma Foundation Fellowship in honor of Robert C. Allen, MD (S.K.), and a generous gift from Dr. Mark J. Daily (J.W.H.). The Bascom Palmer Eye Institute also received funding from NIH Core Grant P30EY014801, Department of Defense Grant #W81XWH-13-1-0048, and a Research to Prevent Blindness Unrestricted Grant.

## Author contributions

M.G.F. analyzed and interpreted the data, and wrote the manuscript. M.A.D. analyzed and interpreted the data, and edited the manuscript. H.A. and L.C. analyzed the data. C. D. collected and prepared tissue samples. A.B. interpreted the data. S.K. wrote the script to plot evolutionary trees, interpreted the data, and edited the manuscript. J.W.H. designed and led the project, provided clinical samples, interpreted the data, and wrote the manuscript.

## Additional information

**Competing interests:** Drs. Harbour and Bowcock are inventors of intellectual property discussed in this study. Dr. Harbour is a paid consultant for Castle Biosciences, licensee of this intellectual property, and he receives royalties from its commercialization. All remaining authors declare no competing financial interests.

