## [Peer Review File · Nature Communications]

Reviewers' comments:

Reviewer #1 (Remarks to the Author):

In this manuscript, Field et al. reconstitute the genomic evolution of uveal melanoma, using the largest series of cases for this rare disease, both from public and in house data. Because of the difficulty to identify BAP1 mutations of high importance for the understanding and prognostication of the disease, they developed a specific bioinformatics pipeline. They also identified potentially novel oncogenic mutations in splicing genes other than SF3B1.

Despite some interesting data in this work, overstatements and experimental weakness impair the quality of the manuscript and need to be addressed.

Major criticisms

- 1) Title is a gross overstatement and does not describe the work. At no point are metastasis or the outcome addressed.
- 2) In the introduction as well as in the discussion, the authors put forward the "novelty" of their discovery by introducing cancer as a progressive Darwinian process. Actually, our understanding of the natural history of cancer has considerably changed in the last 2 years, thanks to the advances in NGS and single cell characterization. Three important papers, cited in the manuscript, have shown the punctuated evolution of cancer genomic alterations in frequent cancers, including colorectal carcinoma, from which the former paradigm of gradual evolution has largely derived.
- 3) The authors mentioned "novel" techniques for their bioinformatics pipeline. What they described is an interesting and probably useful assembly of existing publically available tools. While it allowed for detection of new BAP1 alterations, the number of false indels is lacking and no orthogonal validation is provided. In other words, the sensitivity / specificity of this pipeline are not described.
- 4) The alterations of splicing genes besides SF3B1 are very interesting, but the authors provide no evidence of the causative oncogenic role of these mutations. Almost all splicing mutations are hotspot mutations leading to change-of-function. Supp Table 2 gives only genomic coordinates, which complicates the analysis of the mutations. Truncating mutations of RBM10, SF3A1 and SRSF7 are unlikely to represent such activating mutations. The more convincing recurrent mutation is that of SRSF2, readily available on cbiportal from the TCGA provisional cohort. No functional analysis or splicing analysis is performed.
- 5) The "Mutational signature" brings no further information than that already reported by two (cited) WGS analyses. The authors mentioned the BRCA signature, which is quite imprecise, and do not describe the analysis that was done to exclude it.
- 6) Previous milestone publications analyzed tumor evolution using (i) multiple sampling on different parts of the primary tumor, or of metastases; or (ii) WGS at huge coverage; or (iii) single cell analysis. The authors attempt to reconstitute the natural history of primary UMs using WES at low to fair depth. Furthermore, they focused their analysis on oncogenic mutations, on average two per tumor. Given that, the absence of detected tumor heterogeneity does not rule out its presence, because of the lack of sensitivity of the authors' approach. Furthermore, detected heterogeneity may be due to technical issues: indel frequency is generally underestimated as compared to single nucleotide changes due to less efficient alignment. This would explain the lower BAP1 mutation frequency than the 100% LOH3, (P7L205) in contradiction with LOH3 as secondary hit presented in the introduction (P3L64). An orthogonal validation not sensitive to the type of mutation is mandatory to exclude such bias (TaqMan or ddPCR for example).

Minor criticisms

- 1) In the recent 2016 UICC TNM classification of uveal melanoma, monosomy 3 is the principal molecular prognosis factor in this disease with GEP.
- 2) Intriguingly, some cases present two BSE mutations (AA9A and A985). Are these mutations present in the same tumor cells or part of tumor heterogeneity? These cases should be discussed.
- 3) In the figure 5c, Cancer, not Cancert.

Reviewer #2 (Remarks to the Author):

This is an interesting report that sequences 151 primary UM samples. In addition to the increased detection frequencies of key mutation BAP1 within the patient population, it shows that all signature genomic aberrations including BAP1 occur in an early punctuated burst, which is followed by neutral evolution.

Minor but essential modifications are needed to clarify some issues.

First, this report focuses on the mutation landscape of the primary cancer samples, and there is very limited information about metastatic proclivity. To claim that early punctuated burst of genomic aberrations determines metastatic proclivity is a big stretch, particularly when used in the title. Interestingly, according to the initial report of the two phases of cancer evolution (punctuated or discontinuous vs. stepwise), the changes in phases often involve major evolutionary transitions (including metastasis and drug resistance) (Ref: PubMed 16688757). It is thus interesting to investigate if a new punctuated phase is needed for metastasis in UM. Increased research has linked chromosomal changes to metastasis, for example (Ref: PubMed 27930335). It is also likely that selection on the gene level is neutral following the early punctuated burst, but that strong selection can be detected on the chromosomal level. Is there any useful data to shine light on this issue?

Second, since cancer evolution involves both gene mutation and genome re-organization, it is necessary to point out that one challenge is to integrate the gene mutation landscape with the karyotype landscape, which serves as the genomic context for these gene mutations. In fact, the majority of cancer genome sequencing papers have failed to do so.

Third, authors should spell out more details of the similarities and differences of their results compared with others. For example, have the frequencies of other diverse gene mutations also improved besides BAP1? Are there any surprises which conflict with previous knowledge? What are the rationale/advantages to build the phylogenetic tree based on genomic DNA methylation? Should even larger sample sizes be used for future studies?

Reviewer #3 (Remarks to the Author):

The manuscript describes the analysis of 151 primary uveal melanomas using WES (139), WGS (16), and RNA-seq (~80 TCGA samples). Uveal melanoma is characterized by mutations in Gq signaling pathway genes (GNAQ, GNA11, CYSLTR2 or PLCB4) and "BSE" mutations (BAP1, SF3B1, and EIF1AX)

The authors took care to develop a customized pipeline to improve the analysis of BAP1, and consequently obtain better call rates for mutations in BAP1. This is an important and general insight!

The authors first describe the landscape of driver genes that they detect in uveal melanoma, highlighting the additional BAP1 mutations that they detected that were missed by the standard Broad Institute pipeline. They go on to cluster methylation profiles from TCGA samples, and find (not very surprisingly) that clustering coincides with class and mutation. They then undertake clonality analysis using the Battenberg/Dirichlet prior approach. This analysis reveals the mutational steps in the evolution of uveal melanoma.

It was a pleasure to read this manuscript. It is well written, very clear, quite fascinating, and the

authors have clearly gone to efforts to make the methods reproducible.

The manuscript provides important insights into the order of evolutionary events in uveal melanoma, which is perhaps a simple cancer where the steps are fairly clear but none-the-less interesting.

I recommend that the manuscript be published after the following relatively minor changes are made.

Essential modifications:

Abstract. I felt the opening sentence was too strong, as the idea that cancers may sometimes arise rapidly has been proposed previously. I'd suggest:

"Cancer is thought to generally arise through the gradual accumulation of mutations evolving under Darwinian selection over a long period of time."

A minor issue that should be fixed. Supplementary Table 2 did not convert to a pdf correctly.

The term "deep deletion" is quite awful jargon. I have not come across this much. It's meaning can be inferred but it should be defined clearly on first use.

Figure 1 is quite busy. It does not seem "colorblind" friendly. I'd strongly suggest replacing the array of colored blocks which show class, monosomy/disomy, dataset etc with a present/absent style array, similar to the gene mutation array. It would also be helpful to separately highlight Gq signalling pathway genes and "BSE" genes.

The "Phylogenetic Analysis" sub-section of the results should be renamed to "Unsupervised clustering of methylation data" or something else.

My main criticism is that the raw data is not visible enough. This is probably not feasible for every one of the 151 samples. However, the copy number plots and MAF density plots should be provided for at least the 8 samples for which clonal evolution trees are displayed.

Figure 5. "Gastric cancert (MSI)" should be correted to "Gastric cancer (MSI)"

"A custom python script was developed to automate generation of intra-tumor evolutionary trees, which takes inputs from the above analyses and generates an editable scalable vector graphics (.svg) file visualizing the life history, or phylogenetic tree, for a given sample. This algorithm can be provided upon request." A link to this code should be provided, or it should be made available in github.

Minor suggestions:

It's good to avoid the use of the term 'custom script' in descriptions of bioinformatics pipelines (used twice in the manuscript). It is uninformative and essentially means that that step can be repeated. It would be better to just share the script, where possible.

Response to reviewers' comments:

We wish to thank the reviewers for their extremely helpful comments and suggestions. We have attempted to address each point as thoroughly and thoughtfully as possible. As a result, we believe that the revised manuscript is greatly improved.

Reviewer #1 (Remarks to the Author):

In this manuscript, Field et al. reconstitute the genomic evolution of uveal melanoma, using the largest series of cases for this rare disease, both from public and in house data. Because of the difficulty to identify BAP1 mutations of high importance for the understanding and prognostication of the disease, they developed a specific bioinformatics pipeline. They also identified potentially novel oncogenic mutations in splicing genes other than SF3B1. Despite some interesting data in this work, overstatements and experimental weakness impair the quality of the manuscript and need to be addressed.

Major criticisms

1) Title is a gross overstatement and does not describe the work. At no point are metastasis or the outcome addressed.

RESPONSE: With respect, the canonical mutations (BAP1, SF3B1 and EIF1AX) and chromosome copy number aberrations (monosomy 3, 6p gain and 8q gain) in uveal melanoma are well-attested in the literature to be strongly linked to metastatic risk and, as such, are appropriate to use as surrogate endpoints for metastatic risk and patient outcomes. References 1, 2, 9, 10, 11, 12 and many others in the literature support these statements. Nevertheless, we can see how the title and other wording in the text could be misconstrued to suggest that we studied metastatic tumor samples, which we did not. As such, we have changed the title to “Punctuated evolution of canonical genomic aberrations in uveal melanoma” and have changed wording throughout the Abstract, Introduction and Discussion to clarify this point.

2) In the introduction as well as in the discussion, the authors put forward the “novelty” of their discovery by introducing cancer as a progressive Darwinian process. Actually, our understanding of the natural history of cancer has considerably changed in the last 2 years, thanks to the advances in NGS and single cell characterization. Three important papers, cited in the manuscript, have shown the punctuated evolution of cancer genomic alterations in frequent cancers, including colorectal carcinoma, from which the former paradigm of gradual evolution has largely derived.

RESPONSE: We did not intend to imply that we are the first to describe punctuated evolution in cancer. Indeed, the cancer field is far from having a

comprehensive “understanding of the natural history of cancer,” which is likely to vary considerably among different cancer types. The novelty in this paper is that we are the first to provide evidence for this phenomenon in uveal melanoma, which is particularly interesting since cutaneous melanoma appears to be different. We have re-worded the text to further clarify that our findings are consistent with recent landmark publications and to minimize any confusion (Discussion, 1st paragraph).

3) The authors mentioned “novel” techniques for their bioinformatics pipeline. What they described is an interesting and probably useful assembly of existing publically available tools. While it allowed for detection of new BAP1 alterations, the number of false indels is lacking and no orthogonal validation is provided. In other words, the sensitivity / specificity of this pipeline are not described.

RESPONSE: The modification to our pipeline that is most responsible for our increased BAP1 indel detection is the publically available ABRA (assembly-based indel realigner), which has undergone extensive validation (see Reference #18). Further, most of the samples in our analysis do not have additional DNA available for orthogonal validation because they are from public databases. Nevertheless, we were able to test 5 of our samples that harbor BAP1 indels, 4 with Sanger sequencing and 1 with Ion Torrent. The BAP1 deletion in all 5 samples was orthogonally validated. A formal sensitivity/specificity analysis is beyond the scope of this paper and would not be possible with the currently available materials. Indeed, a formal sensitivity/specificity analysis is the subject of a 5-year 30-center NCI-sponsored trial that we were just funded to perform.

The new validation results are shown below:

Sanger sequencing	MM161 chr3:52,439,812 - 52,439,814	<pre> Fu ACAGAGATCCACACCCACCTACCTGTGTGGTCCCTCAGAGGCTGCAGGGGCC--GTTTGCCTCCAGCACCAGCGGGGACTTGTTCGGCTGATTCAGCTCCAGCAGCTGTGACTCTTGA BAP1 ACAGAGATCCACACCCACCTACCTGTGTGGTCCCTCAGAGGCTGCAGGGGCCCTGTTTGCCTCCAGCACCAGCGGGGACTTGTTCGGCTGATTCAGCTCCAGCAGCTGTGACTCTTGA Consensus ACAGAGATCCACACCCACCTACCTGTGTGGTCCCTCAGAGGCTGCAGGGGCC...GTTTGCCTCCAGCACCAGCGGGGACTTGTTCGGCTGATTCAGCTCCAGCAGCTGTGACTCTTGA </pre>
	MM100 chr3:52,443,747 - 52,443,783	<pre> Fu CCACAGCCCCGATCCGGAGGAGAGAGAGGCTTACCGAATCTCCACAGCAGGAGTGA-----TGTCCAGCAGGCGCTCCCGGACCTCCAGCC BAP1 CCACAGCCCCGATCCGGAGGAGAGAGAGGCTTACCGAATCTCCACAGCAGGAGTGAAGAGGCTGGGTGGGCGCAGAGAGAGGGGGGTGTGTGTCCAGCAGGCGCTCCCGGACCTCCAGCC Consensus CCACAGCCCCGATCCGGAGGAGAGAGAGGCTTACCGAATCTCCACAGCAGGAGTGA.....TGTCCAGCAGGCGCTCCCGGACCTCCAGCC </pre>
	MM66 chr3:52,439,858 - 52,439,860	<pre> Fu CAGAGATCCACACCCACCTACCTGTGTGGTCCCTCAGAGGCTGCAGGGGCCCTGTTTGCCTCCAGCACCAGCGGGGACTTGTTCGGCTGACTTGG--CTCTCAGCAGCTGTGACTCTTGA BAP1 CAGAGATCCACACCCACCTACCTGTGTGGTCCCTCAGAGGCTGCAGGGGCCCTGTTTGCCTCCAGCACCAGCGGGGACTTGTTCGGCTGACTTGG--CTCTCAGCAGCTGTGACTCTTGA Consensus CAGAGATCCACACCCACCTACCTGTGTGGTCCCTCAGAGGCTGCAGGGGCCCTGTTTGCCTCCAGCACCAGCGGGGACTTGTTCGGCTGACTTGG...CTCTCAGCAGCTGTGACTCTTGA </pre>
	MM120 chr3:52,442,599 - 52,442,600	<pre> Fu CCAGGATAGAGACTTCCCGGGACCGGCTCTTCATCATTGAGCAG--AGATATATCCATATACAGGCTGGGGGAGTAGGGGACAGCAGTACCCACAGGGGAGAGAGCAGTACAGCAT BAP1 CCAGGATAGAGACTTCCCGGGACCGGCTCTTCATCATTGAGCAG--AGATATATCCATATACAGGCTGGGGGAGTAGGGGACAGCAGTACCCACAGGGGAGAGAGCAGTACAGCAT Consensus CCAGGATAGAGACTTCCCGGGACCGGCTCTTCATCATTGAGCAG,AGATATATCCATATACAGGCTGGGGGAGTAGGGGACAGCAGTACCCACAGGGGAGAGAGCAGTACAGCAT </pre>
Ion Torrent	MM150 chr3:52,441,141 - 52,441,211	<pre> seq ATACAGAGGCCCCAGCCCCAGCTCCCT-----CAGCCATCCAGCTCAGAGGCCCCCTGT BAP1 ATACAGAGGCCCCAGCCCCAGCTCCCTTCAGAGGATAGCAGAGACCCACAGGCTCCAGCTCATGTGACTACATGATGATGGGATAGACTTCCAGCTCCAGCTCAGAGGCCCCCTGT Consensus ATACAGAGGCCCCAGCCCCAGCTCCCT.....CAGCCATCCAGCTCAGAGGCCCCCTGT </pre>

4) The alterations of splicing genes besides SF3B1 are very interesting, but the authors provide no evidence of the causal oncogenic role of these mutations. Almost all splicing mutations are hotspot mutations leading to change-of-function. Supp Table 2 gives only genomic coordinates, which complicates the analysis of the mutations. Truncating mutations of RBM10, SF3A1 and SRSF7 are unlikely

to represent such activating mutations. The more convincing recurrent mutation is that of SRSF2, readily available on cBioportal from the TCGA provisional cohort. No functional analysis or splicing analysis is performed.

RESPONSE: While the cancer-associated mutations in SF3B1 appear to be change-of-function hotspot mutations (for example, see Reference #20 and PMID: 26842708), this is not the case for all splicing factors. For example, RBM10 undergoes frameshifts, truncations and indels (see PMID: 28091594), similar to what we found in our study. Despite intensive research by many groups investigating a range of cancer types, the causative oncogenic role of splicing gene mutations remains poorly understood and beyond the scope of this paper. Nevertheless, we agree with the reviewer that this finding is interesting and may suggest different oncogenic roles among different splicing factors. We have now included this point in the text (see end of section “Detection of other driver mutations”).

The reviewer does not indicate what format he/she would prefer for describing mutations in Supp Table 2. We believe that genomic coordinates are most appropriate, as they can be used to derive the amino acid change or the cDNA sequence, whereas the opposite is not true. However, we are happy to provide additional information if requested.

5) The “Mutational signature” brings no further information than that already reported by two (cited) WGS analyses. The authors mentioned the BRCA signature, which is quite imprecise, and do not describe the analysis that was done to exclude it.

RESPONSE: The reviewer is correct that mutation signature analysis previously has been reported in uveal melanoma, and we indicate this in the paper. However, we provide new insights, such as the cytosine-to-thymine (C→T) transition in *GNAQ*, *GNA11* and *SF3B1*. As such, we believe that this information is of value. However, we would be happy to remove it at the editor’s request.

6) Previous milestone publications analyzed tumor evolution using (i) multiple sampling on different parts of the primary tumor, or of metastases; or (ii) WGS at huge coverage; or (iii) single cell analysis. The authors attempt to reconstitute the natural history of primary UMs using WES at low to fair depth. Furthermore, they focused their analysis on oncogenic mutations, on average two per tumor. Given that, the absence of detected tumor heterogeneity does not rule out its presence, because of the lack of sensitivity of the authors’ approach. Furthermore, detected heterogeneity may be due to technical issues: indel frequency is generally underestimated as compared to single nucleotide changes due to less efficient alignment. This would explain the lower BAP1 mutation frequency than

the 100% LOH3, (P7L205) in contradiction with LOH3 as secondary hit presented in the introduction (P3L64). An orthogonal validation not sensitive to the type of mutation is mandatory to exclude such bias (TaqMan or ddPCR for example).

RESPONSE: The read depth of the WES and WGS datasets was sufficient for the goal of this study, which was to analyze the clonal relationships between common driver aberrations in a large number of tumor samples. This study is by far the largest and most comprehensive NGS analysis published to date in uveal melanoma. Our goal is very different from those of the “milestone” publications mentioned by the reviewer, which were limited to very small numbers of samples due to the prohibitive cost of those analyses. While there is no doubt that low levels of mostly silent heterogeneity would be detected by deeper sequencing, this would not have impacted our primary goal, which was to analyze the evolutionary relationship between common genomic driver aberrations. Indeed, we show in the WGS samples that low level heterogeneity is due to neutral (silent) tumor evolution and not driver mutations. We did not focus on a small number of oncogenic mutations, rather, we focused on common genomic aberrations, including mutations and chromosomal copy number alterations that are thought to drive tumorigenesis. No publication to date has reported any other frequent driver mutations in uveal melanoma.

We agree with the reviewer that indel frequency can be underestimated relative to simple SNPs. This is why we used Dirichlet modeling and assumed that all mutations in the first cluster are in the most recent common ancestor (MRCA), as we describe in the Methods and Supplementary Fig. 6. Thus, in the few samples with BAP1 indels occurring later than the MRCA, our methodology makes it very unlikely that we would have errantly assigned the mutation to a tumor subclone. Importantly, there were only 5 cases where BAP1 mutations were present in a subclone, and in 2 of these 5 cases the BAP1 mutations were SNPs rather than indels. So, this criticism only applies to only 3 of the 75 cases that were analyzed for clonality. An “orthogonal validation” is not possible since these 3 cases were from the TCGA dataset, and we do not believe that such validation is necessary based on our conservative methodology. Further, since we found in most cases with BAP1 indels that this mutation was present in 100% of tumor cells, under-calling indels does not appear to be an issue using our methodology. Nevertheless, the reviewer makes an important point for the reader to understand, so we have clarified this issue in the Methods, under the “Clonality Analysis” section.

Minor criticisms

1) In the recent 2016 UICC TNM classification of uveal melanoma, monosomy 3 is the principal molecular prognosis factor in this disease with GEP.

RESPONSE: All scientific publications to date comparing GEP to chromosomal markers have shown GEP to be superior in prognostic accuracy.

2) Intriguingly, some cases present two BSE mutations (AA9A and A985). Are these mutations present in the same tumor cells or part of tumor heterogeneity? These cases should be discussed.

RESPONSE: In most cases, these mutations are thought to be present in the same tumor cells because they are both present in 100% of cells. We have now addressed this issue in the text (Discussion, 2nd paragraph).

3) In the figure 5c, Cancer, not Cancert.

RESPONSE: Corrected. Thank you.

Reviewer #2 (Remarks to the Author):

This is an interesting report that sequences 151 primary UM samples. In addition to the increased detection frequencies of key mutation BAP1 within the patient population, it shows that all signature genomic aberrations including BAP1 occur in an early punctuated burst, which is followed by neutral evolution.

Minor but essential modifications are needed to clarify some issues.

First, this report focuses on the mutation landscape of the primary cancer samples, and there is very limited information about metastatic proclivity. To claim that early punctuated burst of genomic aberrations determines metastatic proclivity is a big stretch, particularly when used in the title. Interestingly, according to the initial report of the two phases of cancer evolution (punctuated or discontinuous vs. stepwise), the changes in phases often involve major evolutionary transitions (including metastasis and drug resistance) (Ref: PubMed 16688757). It is thus interesting to investigate if a new punctuated phase is needed for metastasis in UM. Increased research has linked chromosomal changes to metastasis, for example (Ref: PubMed 27930335). It is also likely that selection on the gene level is neutral following the early punctuated burst, but that strong selection can be detected on the chromosomal level. Is there any useful data to shine light on this issue?

RESPONSE: As requested by reviewers 1 and 2, we have changed the title to "Punctuated evolution of canonical genomic aberrations in uveal melanoma." The reviewer raises an intriguing question about the evolution of chromosomal changes in the transition to metastasis. With regard to ongoing selection for

chromosomal alterations in the primary tumor, we did not find any evidence for this. In contrast to cutaneous melanoma, where copy number aberrations tend to occur late, the canonical chromosomal copy number aberrations (CNAs) in uveal melanoma tend to occur early with little ongoing selective pressure. We point this out in the text (Introduction and Discussion). Although an analysis of metastatic samples was beyond the scope of our study, a previous report looking at CNAs showed that there was little evidence for chromosomal evolution between primary and metastatic uveal melanomas (PMID: 19151381).

Second, since cancer evolution involves both gene mutation and genome re-organization, it is necessary to point out that one challenge is to integrate the gene mutation landscape with the karyotype landscape, which serves as the genomic context for these gene mutations. In fact, the majority of cancer genome sequencing papers have failed to do so.

RESPONSE: To our knowledge, the only method that currently exists to determine the evolution of CNAs and mutations in the same region is the Battenberg/Dirichlet methodology that we used. Other clonality algorithms are limited in their methodology as they either 1) only evaluate mutations in regions without CNAs or 2) pair mutations with CNAs in regions with CNAs and assume they occur together. This is a major strength of the methodology used in our paper, and we explain this in the Methods.

Third, authors should spell out more details of the similarities and differences of their results compared with others. For example, have the frequencies of other diverse gene mutations also improved besides BAP1?

RESPONSE: A major advantage of our pipeline is the ability to detect large indels, which usually occur in tumor suppressors like BAP1. Since the other canonical mutations in uveal melanoma are largely point mutations that are easily detected by other pipelines, we did not see an increased frequency of detecting those mutations.

Are there any surprises which conflict with previous knowledge?

RESPONSE: Yes, most previous work suggested that CNAs occur in a successive fashion in uveal melanoma (added as new reference #42). We have included this in the discussion.

What are the rationale/advantages to build the phylogenetic tree based on genomic DNA methylation?

RESPONSE: The rationale for using DNA methylation is that it is agnostic to mutation status and GEP. This is the first such use of methylation data on a large scale in uveal melanoma, and it clearly separates the tumors in an

unsupervised manner that correlates with GEP and mutation status. This point is now explained in the text under heading "Methylomic clustering analysis."

Should even larger sample sizes be used for future studies?

RESPONSE: Yes, and we have now launched an NCI-sponsored 5-year, 30-center prospective trial to obtain larger numbers.

Reviewer #3 (Remarks to the Author):

The manuscript describes the analysis of 151 primary uveal melanomas using WES (139), WGS (16), and RNA-seq (~80 TCGA samples). Uveal melanoma is characterized by mutations in Gq signaling pathway genes (GNAQ, GNA11, CYSLTR2 or PLCB4) and "BSE" mutations (BAP1, SF3B1, and EIF1AX)

The authors took care to develop a customized pipeline to improve the analysis of BAP1, and consequently obtain better call rates for mutations in BAP1. This is an important and general insight!

The authors first describe the landscape of driver genes that they detect in uveal melanoma, highlighting the additional BAP1 mutations that they detected that were missed by the standard Broad Institute pipeline. They go on to cluster methylation profiles from TCGA samples, and find (not very surprisingly) that clustering coincides with class and mutation. They then undertake clonality analysis using the Battenberg/Dirichlet prior approach. This analysis reveals the mutational steps in the evolution of uveal melanoma.

It was a pleasure to read this manuscript. It is well written, very clear, quite fascinating, and the authors have clearly gone to efforts to make the methods reproducible.

The manuscript provides important insights into the order of evolutionary events in uveal melanoma, which is perhaps a simple cancer where the steps are fairly clear but none-the-less interesting.

I recommend that the manuscript be published after the following relatively minor changes are made.

Essential modifications:

Abstract. I felt the opening sentence was too strong, as the idea that cancers

may sometimes arise rapidly has been proposed previously. I'd suggest: "Cancer is thought to generally arise through the gradual accumulation of mutations evolving under Darwinian selection over a long period of time."

RESPONSE: We have changed the opening sentence accordingly.

A minor issue that should be fixed. Supplementary Table 2 did not convert to a pdf correctly.

RESPONSE: We have corrected this. Thank you.

The term "deep deletion" is quite awful jargon. I have not come across this much. It's meaning can be inferred but it should be defined clearly on first use.

RESPONSE: We have changed the term "deep deletion" to "homozygous deletion" throughout the manuscript.

Figure 1 is quite busy. It does not seem "colorblind" friendly. I'd strongly suggest replacing the array of colored blocks which show class, monosomy/disomy, dataset etc with a present/absent style array, similar to the gene mutation array. It would also be helpful to separately highlight Gq signalling pathway genes and "BSE" genes.

RESPONSE: Thank you for the helpful feedback. We have changed the array of colored blocks as requested. We also added colored boxes to demarcate mutations in Gq signaling pathway genes and "BSE" genes, and we added a description in the legend.

The "Phylogenetic Analysis" sub-section of the results should be renamed to "Unsupervised clustering of methylation data" or something else.

RESPONSE: We have renamed this sub-section "Methylomic clustering analysis."

My main criticism is that the raw data is not visible enough. This is probably not feasible for every one of the 151 samples. However, the copy number plots and MAF density plots should be provided for at least the 8 samples for which clonal evolution trees are displayed.

RESPONSE: We have included CNA and MAF plots for these cases in Supplemental Figure 7.

Figure 5. "Gastric concert (MSI)" should be corrected to "Gastric cancer (MSI)"

RESPONSE: Thank you. We have corrected this mistake.

"A custom python script was developed to automate generation of intra-tumor evolutionary trees, which takes inputs from the above analyses and generates an editable scalable vector graphics (.svg) file visualizing the life history, or phylogenetic tree, for a given sample. This algorithm can be provided upon request." A link to this code should be provided, or it should be made available in github.

RESPONSE: We have now made this code publicly available on github (<https://github.com/harbourlab/UPhyloPlot>).

Minor suggestions:

It's good to avoid the use of the term 'custom script' in descriptions of bioinformatics pipelines (used twice in the manuscript). It is uninformative and essentially means that that step can be repeated. It would be better to just share the script, where possible.

RESPONSE: We have removed the term 'custom script' and have made the code publicly available on github (<https://github.com/harbourlab/UPhyloPlot>).

Reviewers' comments:

Reviewer #1 (Remarks to the Author):

This revised manuscript represents a significant improvement compared with the first version. However, a number of caveats and approximations still impair this work.

Comments

- The abstract should describe the precise finding of this work rather than being philosophical or hypothetical.
- We agree with the authors that BAP1 mutations are especially difficult to identify. Their workflow represents a significant advance for clinical diagnostic, but is very similar to most of state-of-the-art variant calling pipelines that use various combinations of variant callers. I understand that sensitivity specificity analysis is very demanding. However this BAP1 pipeline is presented as an advance in the field. So it would be important before implementing it in clinical setting to know its real performance.
- I do not agree with the authors about presenting splicing gene variants as significant mutations. The reader has to go into Supp data to find out the nature and precise consequences of the mutations. I agree that RBM10 and SRSF2 reach some recurrence, and the mutations are compatible with functional consequences. These classic criteria are fully lacking for SF3A1 and SRSF7, and very little evidence justifies displaying them rather than the thousands other non recurrent mutations.
- I still don't agree with the authors about the mutation signature. The prominent C>T signature was published in WGS in Furney et al 4 years ago. What is the value of describing a handful of mutations per sample, furthermore mutations highly driven for functional consequences? The authors use "enriched" (line 164). What are the statistics? I definitively suggest removing this paragraph.
- Indeed, the evolutionary analysis is quite interesting, even if highly predicted by the absence of any recurrent or oncogenic mutations during subclonal evolution of the disease. In this regard, I find the mapping of the putative role of new splicing mutations in subclones quite contradicting the proposed model of neutral evolution (one of the authors' proposals has to be wrong).
- The subclonal mutation of BAP1 in M3, even if rare, would represent a major change in the paradigm of these alterations, i.e. that M3 is enough to transform cells with a Gq mutation. Prior to recent works including this one, a number of M3 UMs were lacking BAP1 mutations, which would fit with the preceding model. However, with refined variant calling (such as presented in this work) ~100% of M3 are associated with BAP1 mutations, making highly unlikely that BAP1 could be subclonal and a secondary events. Therefore, the important finding of the authors has to be very strongly validated, and not only relying on bioinformatics prediction, which, I admit, I am neither mastering, nor trusting. Not surprisingly these subclonal alterations are very large rearrangements difficult to call. Furthermore, numbers of read are much too low to seriously extract reliable frequencies (8 reads only for T3). I understand that the authors are using public data, without possibility to cross validate their finding. However, local realignment of FASTQ data on indel BAP1 seq may allow a better assessment of the allele frequency, and considerably reinforce these data. Without additional evidence, sub-clonal BAP1 story should be discarded.

Reviewer #2 (Remarks to the Author):

I'm satisfied with the authors modifications.

Reviewer #3 (Remarks to the Author):

The authors have now addressed all of my concerns. I recommend the manuscript be published.

Comments from Reviewer #1

The abstract should describe the precise finding of this work rather than being philosophical or hypothetical.

RESPONSE: We have already revised the Abstract to address this reviewer's previous comments, and we believe that it accurately describes the findings of the work. We do not agree that the abstract is "philosophical or hypothetical." We believe that the reviewer's comments reflect a difference of opinion rather than a scientifically relevant issue.

We agree with the authors that BAP1 mutations are especially difficult to identify. Their workflow represents a significant advance for clinical diagnostic, but is very similar to most of state-of-the-art variant calling pipelines that use various combinations of variant callers. I understand that sensitivity specificity analysis is very demanding. However this BAP1 pipeline is presented as an advance in the field. So it would be important before implementing it in clinical setting to know its real performance.

RESPONSE: We present this workflow as a research advance, not as a clinical diagnostic test. It is not being implemented in a clinical setting. Thus, a sensitivity-specificity analysis is beyond the scope of this work, as we stated in our previous response to this reviewer. We attempted to satisfy this reviewer by orthogonally validating all of the complex BAP1 mutations for which we had remaining sample available, but a rigorous sensitivity-specific analysis is not possible at the present time, as the reviewer is surely aware. We have initiated a 5-year, 30-center NCI-sponsored clinical trial, which is the appropriate setting for performing a sensitivity-specificity analysis.

I do not agree with the authors about presenting splicing gene variants as significant mutations. The reader has to go into Supp data to find out the nature and precise consequences of the mutations. I agree that RBM10 and SRSF2 reach some recurrence, and the mutations are compatible with functional consequences. These classic criteria are fully lacking for SF3A1 and SRSF7, and very little evidence justifies displaying them rather than the thousands other non recurrent mutations.

RESPONSE: Although the functional consequences of the SF3A1 and SRSF7 mutations are not known, the tumors containing these mutations cluster with the SF3B1-mutant tumors in the methylomic clustering analysis, despite not harboring SF3B1 mutations (Fig. 3), suggesting that these mutations may confer similar functional consequences as SF3B1 mutations. In addition, SF3A1 and SRSF7 can be found in the same complex with SF3B1, further supporting a functional link between these mutations and SF3B1. Thus, we believe that these are interesting and potentially important mutations that distinguish them from other non-recurrent mutations. We have added clarification of this point to the manuscript.

I still don't agree with the authors about the mutation signature. The prominent C>T signature was published in WGS in Furney et al 4 years ago. What is the value of describing a handful of mutations per sample, furthermore mutations highly driven for

functional consequences? The authors use “enriched” (line 164). What are the statistics? I definitively suggest removing this paragraph.

RESPONSE: We disagree for several reasons. First, the Furney paper contained only 12 cases, whereas our analysis contains 139 cases and includes correlation with molecular subtypes, thereby providing important new information. Second, our findings expand significantly upon those of Furney et al, which did not take flanking bases into account when analyzing mutational signatures, which is the current standard. Thus, they may have identified the C>T signature, but they only looked at whether it was enriched at the 3' position of pyrimidine dimers to see whether it was suggestive of a UV-signature. They concluded that the signature was not enriched in that location, as did we. However, we further analyzed for mutational signatures in the context of both the altered nucleotide base and the flanking bases, and we found a mutational C>T signature specifically at CpG sites, which has been implicated as an aging signature. This was not discovered by Furney et al. Further, it is the very fact that the aging signature was associated with “mutations highly driven for functional consequences” that make it particularly interesting; could it be that mutations associated with aging increase the risk for uveal melanoma? This is of particular interest since advanced age is a risk factor for uveal melanoma.

As the reviewer may not be aware, this type of probabilistic modeling algorithm does not yield conventional statistical significance estimates, such as P-values. Indeed, the methodology used in the Furney paper which the reviewer cites does not provide such estimates either. To provide further clarification, we have re-worded this part of the Results section, and we have further expanded upon our description of the statistical methodology in the Methods section. We also point the reviewer and reader to the reference that clearly describes the mathematical and statistical basis for the mutational signature analysis employed in this study (ref 29).

Indeed, the evolutionary analysis is quite interesting, even if highly predicted by the absence of any recurrent or oncogenic mutations during subclonal evolution of the disease. In this regard, I find the mapping of the putative role of new splicing mutations in subclones quite contradicting the proposed model of neutral evolution (one of the authors' proposals has to be wrong).

RESPONSE: We disagree. We state that “the canonical genomic aberrations in UM usually arise in an early punctuated evolutionary process with little ongoing acquisition of new driver aberrations,” which fits all of our findings. We do not claim that there is no ongoing acquisition of new driver aberrations, which would be highly unlikely. Further, the fact that splicing mutations can be acquired later suggests that they may confer additional survival advantage, which will be an interesting subject of future investigation, especially since a majority of these cases occur in the absence of a BES mutation.

The subclonal mutation of BAP1 in M3, even if rare, would represent a major change in the paradigm of these alterations, i.e. that M3 is enough to transform cells with a Gq mutation. Prior to recent works including this one, a number of M3 UMs were lacking BAP1 mutations, which would fit with the preceding model. However, with refined variant calling (such as presented in this work) ~100% of M3 are associated with BAP1

mutations, making highly unlikely that BAP1 could be subclonal and a secondary events. Therefore, the important finding of the authors has to be very strongly validated, and not only relying on bioinformatics prediction, which, I admit, I am neither mastering, nor trusting. Not surprisingly these subclonal alterations are very large rearrangements difficult to call. Furthermore, numbers of read are much too low to seriously extract reliable frequencies (8 reads only for T3). I understand that the authors are using public data, without possibility to cross validate their finding. However, local realignment of FASTQ data on indel BAP1 seq may allow a better assessment of the allele frequency, and considerably reinforce these data. Without additional evidence, sub-clonal BAP1 story should be discarded.

RESPONSE: We disagree. The reviewer recommends a “local realignment of FASTQ data,” yet one does not use FASTQ files for realignment. If the reviewer meant to say local realignment using the BAM files (the file format of an aligned file), we did that twice. After initial mapping to the human genome, ABRA has a local realignment around potential indels. Additionally, MuTect2 has its own incorporated local realignment that adjusts for tumors with purity less than 100%, multiple subclones, and/or copy number variation (either local or aneuploidy). We have now added text to clarify this point in the Methods. We believe that the analysis we have performed is rigorous and conservative, and that it represents the best available insights into uveal melanoma evolution.

Our findings suggest that monosomy 3 and BAP1 mutation likely occur early and relatively close together in time on a tumor evolutionary scale. However, it is unlikely that these events occur simultaneously, based on known mechanisms of chromosomal loss and gene mutation. Thus, it should not be surprising that one or the other of these events would occasionally be found in a subclone. To delineate early events in UM evolution in greater detail will require an analysis of smaller/earlier tumors (which rarely become available for research sampling) using single cell analysis and other methodologies, which is beyond the scope of this paper.